# Mold-Active Antifungal Prophylaxis in Pediatric Patients with Cancer or Undergoing Hematopoietic Cell Transplantation

**DOI:** 10.3390/jof9030387

**Published:** 2023-03-22

**Authors:** Thomas Lehrnbecher, Konrad Bochennek, Andreas H. Groll

**Affiliations:** 1Division of Pediatric Hematology and Oncology, Hospital for Children and Adolescents, University Hospital, Johann Wolfgang Goethe University, 60589 Frankfurt am Main, Germany; 2Infectious Disease Research Program, Center for Bone Marrow Transplantation, Department of Pediatric Hematology/Oncology, University Children’s Hospital, 48149 Muenster, Germany

**Keywords:** child, cancer, invasive fungal disease, prophylaxis

## Abstract

Invasive fungal diseases (IFDs), in particular invasive mold infections, still pose considerable problems in the care of children and adolescents treated for cancer or undergoing hematopoietic cell transplantation. As these infections are difficult to diagnose, and the outcomes for IFDs are still unsatisfactory, antifungal prophylaxis has become an important strategy in the clinical setting. Antifungal prophylaxis is indicated in patients at high risk for IFD, which is commonly defined as a natural incidence of at least 10%. As there is a growing interest in pediatric-specific clinical trials and pediatric-specific guidelines, this review focuses on the available data of mold-active antifungal prophylaxis in children and adolescents. The data demonstrate that a major effort is needed to characterize the pediatric patient population in which the net effect of prophylactic antifungals will be beneficial as well as to find the optimal prophylactic antifungal compound and dosage.

## 1. Introduction

Invasive fungal diseases (IFDs), in particular invasive mold infections, still pose considerable problems in the care of children and adolescents treated for cancer or undergoing hematopoietic cell transplantation (HCT). These infections are difficult to diagnose, and the outcomes for established IFDs are unsatisfactory. Invasive fungal disease often delays chemotherapy, and a recent study in children treated for acute lymphoblastic leukemia (ALL) showed that IFD is an independent risk factor for a poor outcome [1]. In adult patients, it has been demonstrated that early treatment is associated with s better outcome, which supports the strategy of prophylaxis or empirical therapy [2]. This review focuses on the available data of mold-active antifungal prophylaxis in the pediatric setting, which is still controversially discussed among pediatric hematologists, oncologists and infectious disease specialists.

## 2. Antifungal Strategies

Due to the difficulties to diagnose IFD early as well as due to the high mortality despite adequate treatment, a number of antifungal strategies have been developed (Figure), and each of these strategies has specific advantages and disadvantages [3]. In antifungal prophylaxis, all patients at risk for IFD receive antifungal compounds. According to current guidelines, antifungal prophylaxis is indicated in patients who have a natural incidence of IFD greater than 10% [4,5], and prophylaxis is recommended for patients with acute myeloid leukemia (AML), relapsed acute leukemia, patients undergoing allogeneic HCT, and high-risk ALL. This arbitrary cut-off was set to limit the number-needed-to-treat, as antifungal compounds often exhibit drug interactions and are associated with adverse events (see below). Another approach is the strategy of empirical therapy, which is established in febrile neutropenic patients at high risk for IFD (e.g., expected time of neutropenia ≥ 10 days) who do not respond to broad-spectrum antibiotics after 72 to 96 h or who have recurrence of fever (Figure 1). Empirical therapy can be considered either as prophylaxis for patients at the highest risk for IFD, or as treatment of an infection that is still occult. The feasibility and efficacy of empirical therapy has been assessed in large studies in adults, and results were confirmed in the pediatric setting [6,7,8]. To further limit the number-needed-to-treat, some centers use the pre-emptive approach. In this strategy, antifungal therapy is established only in those neutropenic febrile patients who (1) have not responded to broad-spectrum antibiotics (as in empirical therapy) and (2) additionally, have suggestive infiltrates in the CT scan and/or a positive galactomannan test indicating IFD (Figure 1). A recent clinical trial in adults on fluconazole prophylaxis confirmed that, compared to empirical therapy, pre-emptive treatment is not associated with higher mortality but can safely spare mold-active antifungal compounds [9]. In the pediatric setting, there has only been one randomized trial comparing empirical with pre-emptive therapy, which confirmed the results in adult patients [10]. A limiting factor of the pre-emptive approach is the fact that both a CT scan and the results of the galactomannan assay have to be rapidly available. Whereas the performance of the galactomannan assay is similar in the pediatric and adult setting, there are conflicting results on the occurrence of typical signs of invasive aspergillosis in imaging studies in children (e.g., halo sign, air crescent sign) [11,12,13]. The principles of treatment of established IFDs, which are categorized according to host factors, and clinical and microbiological findings in possible, probable and proven IFD, respectively, do not differ between children and adults [4,14]. These principles include the facts that treatment should be started as early as possible, that resistance testing of the pathogen should be performed whenever possible, and that predisposing factors should be controlled (e.g., immunosuppression). 

## 3. Risk Factors for Invasive Fungal Disease and Epidemiology

A systematic review demonstrated that prolonged neutropenia, high-dose steroid exposure, intensive chemotherapy for AML, and acute and chronic graft-versus-host disease (GvHD) are important risk factors for IFD in pediatric cancer patients [15]. Unfortunately, no threshold for the steroid dose could be determined. 

In the clinical trial CCG 2961, which enrolled 492 children with AML, 18%, 21%, and 14% of children experienced an IFD in treatment phase 1, 2, and 3, respectively [16]. The incidence of mold and yeast infection was similar, and 31% and 25.9% of infection deaths were associated with *Aspergillus* spp. and *Candida* spp., respectively. Another analysis of 35 and 31 children treated in Thailand for AML and relapsed leukemia reported an incidence of 11.4% and 19.3% of IFD, respectively [17]. Notably, antifungal prophylaxis was not regularly prescribed in these patients and often consisted of fluconazole. The situation is even more complicated in children with ALL, the largest patient population in pediatric hematology and oncology. Overall, most studies report an incidence of IFD lower than 5% [1,18]. However, an analysis of the multi-national, multi-institutional clinical trial AIEOP-BFM 2009 demonstrated that certain subgroups of patients have a risk for IFD of 10% and higher, such as patients older than 12 years or those with insufficient treatment response on day 15 [1]. Unfortunately, there was no information on the use of antifungal prophylaxis, which was explained by the facts that it is unclear to date whether antifungal prophylaxis has any significant effect in patients with ALL, and that widely used prophylactic approaches such as extended dosing regimens of compounds such as liposomal amphotericin B (L-AmB) or micafungin have not demonstrated significant antifungal efficacy in any randomized trial. In pediatric patients treated for sarcoma, IFDs are rarely seen [19]. 

In the setting of pediatric allogeneic HCT, a retrospective single-center study of 209 children transplanted between 2004 and 2012 reported an incidence of IFD of 12% despite antifungal prophylaxis with fluconazole (until 2008) and voriconazole (after 2008), respectively [20]. Importantly, patients who developed IFD had a significantly increased risk of treatment-related mortality (OR 3.773, *p* = 0.004).

## 4. Mold-Active Antifungal Compounds for Prophylaxis

Among mold-active antifungal compounds, voriconazole (for patients ≥ 2 years of age), posaconazole (for patients ≥ 2 years of age), and micafungin (for prophylaxis against invasive *Candida* infections) are currently approved for prophylactic use in the pediatric setting. Both voriconazole and posaconazole are involved in the CYP450 pathway and exhibit a number of important drug interactions with a variety of compounds including cyclosporine, tacrolimus, and erythromycin. Importantly, the concomitant use of an azole with vinca alkaloids such as vincristine, which is a cornerstone of the treatment of ALL, is contraindicated due to neurotoxicity [21]. The advantage of azoles is the fact that these drugs are available as both oral and intravenous formulations, whereas the echinocandins as well as amphotericin B formulations need to be administered intravenously. For voriconazole, posaconazole, and micafungin, specific dosages for the prophylactic use have been recommended, whereas this is unclear for caspofungin or amphotericin B, which are widely used in this setting. In addition, although not approved, a number of experts use extended dosing regimens, which facilitates the use in the prophylactic setting [22].

## 5. Methods

We searched the PubMed database for articles on antifungal prophylaxis in children with cancer or undergoing HCT (Table 1). The main search terms were “pediatric OR children; hematology OR cancer OR leukemia OR hematopoietic cell transplantation; antifungal prophylaxis”. Retrieved references were screened by hand for additional references. References with mixed pediatric and adult patients were only included if sufficient information on the specific results in children and adolescents was available or the proportion of pediatric patients allowed a solid conclusion.

## 6. Clinical Trials Investigating Prophylactic Antifungal Compounds in Homogenous Patient Populations

### 6.1. Voriconazole in Patients with Acute Leukemia and in Patients Undergoing HCT

A randomized study compared prophylactic voriconazole (6 mg/kg/dose for initial 2 doses, then 4 mg/kg per dose) to low dose amphotericin B (0.5 mg/kg thrice weekly) in children with AML (*n* = 30) or ALL (*n* = 70) [23]. The failure of prophylaxis was defined as a composite endpoint and occurred in 14 out of 50 patients (28%) in the voriconazole arm and 17 out of 50 patients (34%) in the amphotericin B arm (*p* = 0.66). Specifically, one proven mucormycosis and one possible IFD were diagnosed in children receiving voriconazole, and three possible IFDs in children on amphotericin B prophylaxis. Eleven children on voriconazole and 13 children on amphotericin B were switched to empirical antifungal therapy, and prophylaxis was stopped prematurely in one patient in each arm. Drug-related serious adverse events were seen in 6% and 30% in the voriconazole and amphotericin B arm, respectively (*p* < 0.01). The authors concluded that voriconazole and amphotericin B are similarly effective, but voriconazole is less toxic than amphotericin B. 

Maron et al. analyzed the incidence, epidemiology, and outcome of IFD in two cohorts of children treated for AML (105 patients treated before 2002 in the trial AML97, 117 patients treated after 2002 in the trial AML02). Voriconazole prophylaxis was given to the AML02 cohort at a dosage of 200 mg twice per day for patients weighing 40 kg or more and of 100 mg twice per day for children weighing less than 40 kg. A voriconazole dose of 4 mg/kg every 12 h was chosen if the drug had to be administered intravenously. Unfortunately, the authors did not specify antifungal prophylaxis in the AML97 cohort. The incidence of IFD was similar in both groups. Twelve (11.4%) patients in AML97 and 11 (9.4%) patients in AML02 developed IFD. However, the authors observed a significant shift in the pathogens. Whereas IFD in AML97 was mainly due to *Aspergillus* spp. (*Aspergillus flavus* (*n* = 7), *Aspergillus fumigatus* (3), *Fusarium oxysporum* (1), *Rhizopus* spp. (1), undefined (1)), fungal infections in AML02 were mainly caused by phaeohyphomycoses (*Alternaria* spp. (3), *Curvularia* spp. (2), *Exserohilum rostratum* (2), *Fusarium* spp. (2), *Bipolaris* spp. (1), *Macrophomina phaseolina* (1), *Mucor* spp. (1), *Pseudallescheria boydii* (1), *Rhizopus* spp. (1)). Importantly, mortality attributable to IFD after 90 days was significantly lower in those patients who received voriconazole prophylaxis (none of 105 patients in the AML02 group) compared with those who did not receive voriconazole prophylaxis (4 of 117 patients in the AML97 group) (*p* = 0.05). The authors concluded that voriconazole prophylaxis decreases the mortality of IFD but also leads to a marked shift in the pattern of pathogens.

Voriconazole was also studied in 56 children (mean age 9.05 years, range 2–17) after allogeneic HCT [24]. In 23 patients, voriconazole was given at a dosage of 5 mg/kg per 12 h, and in 33 children at a dosage of 7 mg/kg per 12 h with a limiting dose of 200 mg per 12 h. Prophylaxis was administered from day +1 until day +75 in patients suffering from GvHD even longer. Within the follow-up period of 6 months, one patient (1.8%) developed a fatal IFD, classified as probable aspergillosis, and nine children (16.1.%) required either empirical (*n* = 7) or pre-emptive (*n* = 2) treatment. In 10 (17.8%) patients, voriconazole prophylaxis had to be stopped prematurely due to adverse events (median, day +26.5), which mostly occurred as elevated liver enzymes. The authors concluded that voriconazole is a safe and effective antifungal prophylaxis in children after allogeneic HCT.

### 6.2. Posaconazole in Patients Undergoing HCT

A retrospective single-center study analyzed the efficacy of posaconazole suspension (31 children; 16 patients younger than 6 years) and posaconazole tablets (32 children; 19 patients older than 13 years) given as antifungal prophylaxis after allogeneic HCT [25]. The suspension was administered at a dosage of 4 mg/kg three times daily, and the tablets at a dosage of 5–7 mg/kg twice on day 1, followed by 5–7 mg/kg once daily. Within the median observation period of 107 days, none of the patients in either group developed proven, probable, or possible IFD. Posaconazole trough levels were significantly higher in patients receiving tablets than in those with the suspension. The authors conclude that posaconazole tablets are safe and effective in children after HCT.

### 6.3. Caspofungin in Patients with AML or in Patients Undergoing Allogeneic HCT

One multi-center, randomized, open-label study in patients aged 3 months to 30 years suffering from AML was recently published by Fisher et al. [26]. Patients were treated in 115 US and Canadian institutions between April 2011 and November 2016 and were randomly assigned during the first chemotherapy cycle to prophylaxis with caspofungin (*n* = 257) or fluconazole (*n* = 260). Prophylactic caspofungin was given at a dosage of 70 mg/m^2^ on day 1 (maximum dose 70 mg) followed by 50 mg/m^2^ per day (maximum dose 50 mg per day) during neutropenia following each cycle of chemotherapy. The clinical trial was terminated early after enrolling 517 participants with a median age of 9 years (range, 0–26 years) as the second interim analysis suggested futility. The primary outcome of the study was the occurrence of IFD, and a total of 23 proven or probable IFDs were observed. Seventeen infections occurred in the fluconazole arm, and six in the caspofungin arm. The 5-month cumulative incidence of proven or probable invasive fungal disease was 3.1% (95% CI, 1.3–7.0%) in the caspofungin arm compared to 7.2% (95% CI, 4.4–11.8%) in the fluconazole arm, which was a significant difference (*p* = 0.03 by log-rank test). Notably, the difference between the caspofungin and fluconazole arm was also significant for proven or probable invasive aspergillosis, which was a secondary endpoint of the study (0.5% (95% CI, 0.1–3.5%) in the caspofungin arm versus 3.1% (95% CI, 1.4–6.9%) in the fluconazole arm (*p* = 0.046 by log-rank test)). No statistically significant differences in the administration of empirical antifungal therapy and overall survival were seen. This trial shows that antifungal prophylaxis with caspofungin significantly reduces the incidence of both invasive fungal disease and of invasive aspergillosis in children, adolescents, and young adults with AML. 

In a retrospective monocenter study in children under the age of 18 years undergoing allogeneic HCT, caspofungin given in the prophylactic setting was compared to L-AmB given at a dosage of 1 mg/kg daily [27]. Liposomal amphotericin B was used as prophylaxis prior to 2008, and due to a high incidence of nephrotoxicity, antifungal prophylaxis was switched to caspofungin after 2008. Sixty patients each received L-AmB (median age, 7.5 years) at a dosage of 1 mg/kg per day, or caspofungin (median age, 9.5 years) at a dosage of 50 mg/m^2^ per day (maximum dose 50 mg per day). No proven breakthrough fungal disease occurred in either group, and one patient receiving caspofungin prophylaxis developed probable invasive aspergillosis. Liposomal amphotericin B had more drug-related adverse events, in particular hypokalemia. The authors concluded that caspofungin and L-AmB exhibit similar activity in the prophylactic setting in allogeneic HCT but that L-AmB is associated with more adverse events. 

A multi-center, randomized, open-label trial compared the prophylactic use of caspofungin (70 mg/m^2^ (maximum 70 mg) on day 1, followed by 50 mg/m^2^ (maximum 50 mg) daily) with a center-specific triazole, which was either fluconazole or voriconazole in children and adolescents undergoing allogeneic HCT [28]. Antifungal prophylaxis was given from day 0 until day +42 or discharge. Overall, 290 patients (median age, 9.5 years (range 0.3–20.7)) were randomized to receive caspofungin (*n* = 144) or a triazole (*n* = 146; fluconazole, *n* = 100; voriconazole, *n* = 46). The day 42 cumulative incidence of proven or probable IFD was 1.4% (95% CI, 0.3–5.4%) in the caspofungin group compared to 1.4% (95% CI, 0.4–5.5%) in the triazole group (*p* = 0.99, log-rank test). Further analysis did not show any significant difference in the occurrence of proven or probable IFD between caspofungin and fluconazole (*p* = 0.78) or voriconazole (*p* = 0.69), respectively. Due to the overall low incidence of IFD, the trial was prematurely closed. 

### 6.4. Micafungin in Patients with ALL or in Patients Undergoing Allogeneic HCT

The administration of micafungin administered at a dosage of 9 mg/kg twice weekly was evaluated in 61 pediatric patients undergoing induction therapy for ALL [29]. This study primarily analyzed pharmacokinetics and found that the proposed regimen seems to result in adequate exposure for *Candida* therapy. Although the authors stated that the generalizability of the results for *Aspergillus* prophylaxis could not be provided without assumptions on target concentrations, the clinical experience using this dosage was reported in abstract form [30]. A total of 169 children (median age, 4 years) suffering from ALL received micafungin prophylaxis during the first 5 weeks of treatment, when azoles are contraindicated due to the concomitant use of vincristine. This cohort was compared to historical control patients (*n* = 643; median age, 5 years) who did not receive antifungal prophylaxis during the induction course. Two of the 169 patients (1.2%) receiving micafungin developed proven or probable invasive aspergillosis, whereas this was seen in 36 out of the 643 patients (5.6%) of the historical control (*p* = 0.013). However, one has to note that the dose of micafungin given in this trial is considerably higher than that approved for prophylaxis or therapy. 

One study evaluated safety and efficacy in 38 children (median age (range), 7.3 years (0.4–18.7)) undergoing HCT [31]. Micafungin was given prophylactically at a dosage of 2 mg/kg per day. Probable invasive aspergillosis was diagnosed in one patient on day +59, and possible IFD in another patient on day +120. Both patients recovered with a combination therapy with L-AmB and itraconazole or L-AmB monotherapy, respectively. Notably, in none of the three patients who had suffered from IFD prior to HCT, a breakthrough infection occurred. The authors concluded that micafungin prophylaxis is safe and feasible in pediatric allogeneic HCT.

## 7. Clinical Trials Investigating Prophylactic Antifungal Compounds in Heterogenous Patient Populations

### 7.1. L-AmB at Different Dose Schedules in Children at High Risk for IFD

In a single-center study, 46 children (mean age, 7.7 years) with underlying malignancies such as ALL, AML, or relapsed acute leukemia were considered at high risk for IFD (median duration of neutropenia (<500/µL), 10 days) [32]. These patients received antifungal prophylaxis with L-AmB at a dosage of 2.5 mg/kg twice weekly. None of these patients developed proven or probable IFD, which was significantly less than a comparable historical control group of 45 patients without antifungal prophylaxis, in whom five proven and two probable IFDs were diagnosed (*p* = 0.01). No impact of L-AmB prophylaxis was seen on the use of empirical antifungal therapy. Prophylaxis with L-AmB had to be discontinued in four patients (8.7%) due to allergic reactions. Hypokalemia of less than 3.0 mmol/L was seen in 13.5% of the 187 prophylactic episodes. The authors concluded that L-AmB given twice weekly in children at high risk for IFD is safe and seems to be effective. 

In a single-center study, 19 patients (16 of them treated for ALL, median age 6.5 years) received L-AmB as antifungal prophylaxis at a dosage of 10 mg/kg once weekly [33]. L-AmB was given as a 2 h infusion, and the study period was 3 months (median, 6 infusions per patient, range 1–12 infusions). Five patients (26%) suffered from infusion-related reactions, one patient met the criteria of transient nephrotoxicity, and seven patients (37%) experienced hypokalemia. The authors concluded that L-AmB prophylaxis at a dosage of 10 mg/kg once a week is poorly tolerated. 

Different results were reported by a small prospective case series, in which L-AmB 10 mg/kg was given once weekly as secondary prophylaxis to children (median age, 8.5 years) who required further immunosuppressive therapy after treatment for IFD [34]. In none of the patients was the secondary prophylaxis (median number of administrations, 17, range, 1–58) prematurely stopped due to adverse events. In six patients, no relapse of IFD occurred, whereas in one patient, IFD worsened despite secondary prophylaxis.

A retrospective single-center study compared 116 children (median age 6.1 years) who were treated for ALL or AML until 2010 and who mostly received fluconazole as antifungal prophylaxis with 84 patients (median age, 4.69 years) treated for acute leukemia after 2010, in whom L-AmB was given as prophylaxis at a dosage between 3 and 5 mg/kg three times weekly [35]. In the historical control group with fluconazole prophylaxis, 10 patients developed proven IFD, mostly invasive aspergillosis, whereas in the patients receiving L-AmB, no patients were diagnosed with IFD (*p* = 0.007). In addition, vincristine-induced neurotoxicity was significantly less often observed in patients with L-AmB compared to those receiving fluconazole. 

### 7.2. Micafungin in Children at High Risk for IFD

A single-center retrospective study included 21 children (median age, 9 years) at high risk for IFD such as patients treated for high-risk ALL, AML, or leukemia relapse [36]. Patients received micafungin at a dose of 3 to 4 mg/kg twice weekly as antifungal prophylaxis. Proven or probable breakthrough invasive fungal disease was not observed in any patient. Notably, plasma micafungin trough concentrations exceeded 150 ng/mL, a concentration proposed to be effective for prophylaxis. The authors concluded that micafungin at 3 to 4 mg/kg given twice weekly could be a safe and efficient alternative for antifungal prophylaxis in children at high risk for IFD.

## 8. Critical Summary of Studies on Antifungal Prophylaxis in Children

The ECIL8 recommendations on the use of antifungal compounds in the pediatric setting are based on four components, namely the evidence for efficacy from adult phase 2 and 3 trials, the existence and quality of pediatric pharmacokinetic data and dosing recommendations, specific pediatric safety data and supportive efficacy data, and regulatory approval for use in pediatric age groups by the European Medicines Agency (EMA) [4]. These considerations already reflect the limitations of the recommendations of antifungal prophylaxis in the pediatric setting for a number of reasons. First, the evidence for efficacy in prophylaxis may not be transferred from adult patients, as pediatric malignancies and treatment protocols are different from those in adults. Second, there are a number of differences between children and adults, such as co-morbidities or the speed of immunoreconstitution after chemotherapy or allogeneic HCT, all of which impact on the risk of IFD [37]. Finally, the regulatory approval of antifungal compounds is often lacking in the pediatric setting, in particular for children below the age of 2 years. 

The fact that only three studies were performed in a randomized way and that many trials were underpowered additionally reflects specific problems of clinical trials in the pediatric setting, but could be, at least in part, explained by the small number of children with cancer compared to adults. Furthermore, most studies were of retrospective nature, and historical controls were often used as comparators, which may be associated with major bias. Whether dosages outside the approved dose recommendations need to be studied in these trials is at least questionable.

Despite these limitations, one has to highlight that there is a growing interest in pediatric-specific clinical trials and pediatric-specific guidelines. In addition, all applications for marketing authorization for new drugs now have to include the results of studies as described in an agreed pediatric investigation plan (PIP), which is a development plan in order to obtain data through studies in children. This development is encouraging, although it is clear that a major effort is needed to characterize the pediatric patient population in which the net effect of prophylactic antifungals will be beneficial as well as to find the optimal prophylactic antifungal compound and dosage.

## Figures and Tables

**Figure 1 jof-09-00387-f001:**
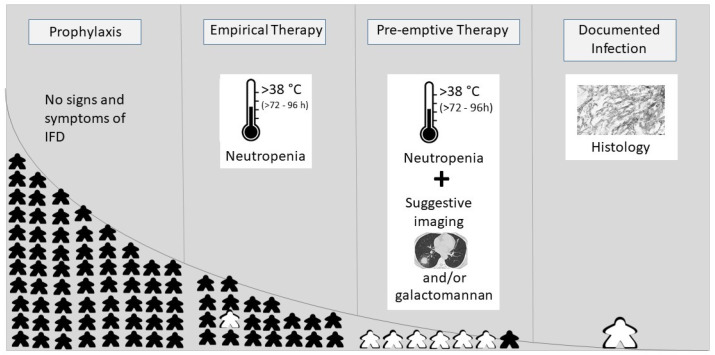
Antifungal strategies and number-needed-to-treat. The number-needed-to-treat is represented by 
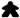
, with black as “not infected” and white as “infected”; IFD, invasive fungal disease.

**Table 1 jof-09-00387-t001:** Overview of clinical trials on antifungal prophylaxis in pediatric patients with cancer or undergoing hematopoietic cell transplantation. IFD, invasive fungal disease; IA, invasive aspergillosis; ALL, acute lymphoblastic leukemia; AML, acute myeloid leukemia; HCT, hematopoietic cell transplantation; L-AmB, liposomal amphotericin B.

Authors (Ref)	Year	Study Design	Study Population (n)	Antifungal Agents	Authors’ Conlcusion
Bury et al.	2022	prospective	ALL induction (63)	micafungin 9 mg/kg 2×/week	adequate exposure to *Candida* infection, no statement for *Aspergillus* possible
Bury et al. (abstract)	2022	prospective, historical control	ALL induction (812)	micafungin 9 mg/kg 2×/week	micafungin decreases incidence of IA in ALL induction
Dvorak et al.	2021	randomized, open label	allogeneic HCT (290)	caspofungin, fluconazole, voriconazole	no difference in efficacy between caspofungin, fluconazole and voriconazole
Fisher et al.	2019	randomized open label	AML (517)	caspofungin, fluconazole	caspofungin significantly reduces incidence of both IFD and of IA
Döring et al.	2017	retrospective, single center	HCT (63)	posaconazole tablets and suspension	both posaconazole tablets and suspension are effcetive after HCT
Bochennek et al.	2015	retrospective, single center	mixed population (21)	micafungin 3–4 mg/kg 2×/week	micafungin seems to be feasible for antifungal prophylaxis
Yoshikawa et al.	2014	prospective, uncontrolled	allogeneic HCT (38)	micafungin	micafungin is safe and feasible
Hand et al.	2014	prospective, single center	mostly ALL (19)	LAmB 10 mg/kg once weekly	high-dose L-AmB poorly tolerated
Maron et al.	2013	retrospective, single center	AML (213)	voriconazole	number of IFD unchanged, but shift in pathogens
Döring et al.	2012	retrospective, historical control	allogeneic HCT (120)	caspofungin, L-AmB	caspofungin and L-AmB with similar activity, L-AmB with more adverese events
Molina et al.	2012	retrospective, single center	HCT (56)	voriconazole, low dose amphoteriicn B	voriconazole safe and effective in HCT
Ginocchio et al.	2012	prospective, single center	mixed population, secondary prophylaxis (7)	LAmB 10 mg/kg once weekly	high-dose L-AmB tolerable
Mandhaniya et al.	2011	randomized, open label	acute leukemia (100)	voriconazole, low dose amphoteriicn B	voriconazole and amphotericin B similarly effectve, with less toxicity of voriconazole
Bochennek et al.	2011	prospective, historical control	mixed population (83)	LAmB 2.5 mg/kg 2×/week	L-AmB safe, seems to decrease risk for IFD

## Data Availability

Not applicable.

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
