# Peer review of "Mold-Active Antifungal Prophylaxis in Pediatric Patients with Cancer or Undergoing Hematopoietic Cell Transplantation"

_jof, 2023, doi:10.3390/jof9030387_

Round 1
Reviewer 1 Report
The authors review the debated scenario of antifungal prophylaxis in the pediatric cancer setting. This is a very interesting item, since invasive fungal diseases are often serious and life threatening infections.
General comment: after a very interesting presentation of the current literature on mold prophylaxis, the conclusions should stress, to be clear for all readers, that the published consensus and guidelines on this topic identified risk patients who benefy from antifungal prophylaxis, however the need to better characterize such risk population is still an open issue. Otherwise the last sentence of the abstract repeated at the end of the paper might be misleading.
"The data demonstrate that major effort is needed to characterize the pediatric patient population in which the net effect of prophylactic antifungals will be benficial..."
Moreover in the paragraph 3 about risk factors I would add a brief comment on evaluation of local epidemiology and patient previous history to better evaluate patient' risk profile for IFI, as reported in the published "8th European Conference on Infections in Leukaemia" guidelines.
Line 96: I believe that the sentence on Pediatric sarcomas sounds out of place, I would suggest to delete it together with the reference 19
"In pediatric patients treated for sarcoma, IFD are rarely seen [19]."Line 199: Seventeen percent of these infections occurred in the fluconazole arm, six in the caspofungin arm.
The numbers seventeen and six should be absolute numbers, and not percentages according to the original paper (Fisher et al JAMA 2019).
Typing mistakes: abstract, line 20 beneficial, the "e" is missing; line 113: micafungin: the a "a" is missing
Author Response
We thank the reviewer for the thoughtful review and the positive comments, The authors review the debated scenario of antifungal prophylaxis in the pediatric cancer setting. This is a very interesting item, since invasive fungal diseases are often serious and life threatening infections. Thank you for this positive comment. General comment: after a very interesting presentation of the current literature on mold prophylaxis, the conclusions should stress, to be clear for all readers, that the published consensus and guidelines on this topic identified risk patients who benefy from antifungal prophylaxis, however the need to better characterize such risk population is still an open issue. Otherwise the last sentence of the abstract repeated at the end of the paper might be misleading. "The data demonstrate that major effort is needed to characterize the pediatric patient population in which the net effect of prophylactic antifungals will be benficial..." We agree with the reviewer that the last sentence is somehow misleading. We modified the sentence to “……major effort is needed to better characterize the pediatric patient population” and hope that the reviewer agrees that this will clarify the statement. Moreover in the paragraph 3 about risk factors I would add a brief comment on evaluation of local epidemiology and patient previous history to better evaluate patient' risk profile for IFI, as reported in the published "8th European Conference on Infections in Leukaemia" guidelines. We fully agree with the reviewer and included a statement on local epidemiology and previous IFD in the risk of IFD (“It is important to note that a previous IFD as well as the local epidemiology also have an impact on the individual risk for IFD and have to be taken into account in the decision whether a patient should receive antifungal prophylaxis or not.“). Line 96: I believe that the sentence on Pediatric sarcomas sounds out of place, I would suggest to delete it together with the reference 19 "In pediatric patients treated for sarcoma, IFD are rarely seen [19]." On one hand we agree with the reviewer´s comment, on the other hand we thought that this statement is important to underline that no antifungal prophylaxis should be routinely given in any patient suffering from a solid tumor. If the reviewer accepts we would like to keep this statement in the manuscript, but added “and therefore, these patient should not routinely receive antifungal prophylaxis” for clarification. Line 199: Seventeen percent of these infections occurred in the fluconazole arm, six in the caspofungin arm. The numbers seventeen and six should be absolute numbers, and not percentages according to the original paper (Fisher et al JAMA 2019). We have corrected the errors. Typing mistakes: abstract, line 20 beneficial, the "e" is missing; line 113: micafungin: the a "a" is missing We thank the reviewer for pointing out the spelling errors, which we have corrected.Reviewer 2 Report
Invasive fungal diseases for children cancer patients deserve clinic research as these infections are difficult to diagnose. This review summarized the prophylactic antifungal compounds and dosing. Some issues should be concerned before its acceptance.
1. How many children patients suffer from the IFD?
2. Only PUBMED database was screened, it may be not sufficient
3. The environment issue should be considered. Children from different contourites will face different diagnosis and treatment level, fungal diseases also have many relationship with the place they live in.
Author Response
Invasive fungal diseases for children cancer patients deserve clinic research as these infections are difficult to diagnose. This review summarized the prophylactic antifungal compounds and dosing. Some issues should be concerned before its acceptance. 1. How many children patients suffer from the IFD? As we have stated several times in the manuscript, IFDs are difficult to diagnose. In addition, as stated as well in the original manuscript, the incidence of IFD depends on the different risk groups. Therefore, we are unable to give the number of children who suffer from IFD. However, we are not sure whether we might have misunderstood the reviewer´s question. 2. Only PUBMED database was screened, it may be not sufficient We agree with the reviewer´s criticism. We therefore have included in the original manuscript the statement that “Retrieved references were screened by hand for additional references”. It is also important to note that we did no present a systematic review, and therefore, we did not run additional searches on other platforms. 3. The environment issue should be considered. Children from different contourites will face different diagnosis and treatment level, fungal diseases also have many relationship with the place they live in We fully agree with the reviewer that this statement was missing (see also criticism by reviwer #1). We therefore included a statement on this at the end of the first paragraph of section 3 (It is important to note that a previous IFD as well as the local epidemiology also have an impact on the individual risk for IFD and have to be taken into account in the decision whether a patient should receive antifungal prophylaxis or not [4]).